# The Incidence Trend and Management of Thyroid Cancer—What Has Changed in the Past Years: Own Experience and Literature Review

**DOI:** 10.3390/cancers15204941

**Published:** 2023-10-11

**Authors:** Krzysztof Kaliszewski, Dorota Diakowska, Michał Miciak, Krzysztof Jurkiewicz, Michał Kisiel, Szymon Makles, Anna Dziekiewicz, Szymon Biernat, Maksymilian Ludwig, Bartłomiej Ludwig, Karolina Sutkowska-Stępień, Maciej Sebastian, Paweł Domosławski, Krzysztof Sutkowski, Beata Wojtczak

**Affiliations:** 1Department of General, Minimally Invasive and Endocrine Surgery, Wroclaw Medical University, 50-367 Wrocław, Poland; michal.miciak@student.umw.edu.pl (M.M.); krzysztof.jurkiewicz@student.umw.edu.pl (K.J.); michal.kisiel@student.umw.edu.pl (M.K.); szymon.makles@student.umw.edu.pl (S.M.); anna.dziekiewicz@student.umw.edu.pl (A.D.); szymon.biernat@student.umw.edu.pl (S.B.); bartek.ludwig@onet.pl (B.L.); karolina.sutkowska@onet.pl (K.S.-S.); maciej.sebastian@umw.edu.pl (M.S.); pawel.domoslawski@umw.edu.pl (P.D.); krzysztof.sutkowski@umw.edu.pl (K.S.); beata.wojtczak@umw.edu.pl (B.W.); 2Department of Basic Sciences, Faculty of Health Science, Wroclaw Medical University, 50-367 Wrocław, Poland; dorota.diakowska@umw.edu.pl

**Keywords:** thyroid cancer management, incidence trends, NIFTP, PTC, personalized medicine, COVID-19, cancer screening activity

## Abstract

**Simple Summary:**

The incidence of thyroid cancer (TC) has increased greatly since the 1970s, mostly because of higher detection of the small forms of papillary thyroid cancer (PTC). Despite this main and popular opinion, an increase in the occurrence of the larger forms of TC has also been observed. Due to some noticed trends concerning TC incidence and behavior, the management of these neoplasms has also changed in recent years. “Cancer screening activity”, which is the most popular of the aforementioned events, might have increased the PTC number during the analyzed period. However, the observed increase in the occurrence of TC continued to a point in time where there was a slowdown and even a reversal. This phenomenon is still debated and worthy of further investigation.

**Abstract:**

Because of ambiguous and widely debated observations concerning the incidence, trend, and management of TC, we performed this analysis. We drew attention to some events, such as “cancer screening activity”, introduction of noninvasive follicular neoplasm with papillary-like nuclear features (NIFTP) to TC types, possibility of papillary thyroid microcarcinoma (PTMC) active surveillance (AS), occurrence of personalized medicine in TC management, and, finally, COVID-19 pandemic time. Because of the opinion that all changes have been made mostly by PTC, we compared it to the remaining types of TC in terms of incidence, clinical and pathological characteristics, and treatment. We analyzed patients treated in a single surgical center in eastern Europe (Poland). The prevalence of TC significantly increased from 5.15% in 2008 to 13.84% in 2015, and then significantly decreased to 1.33% in 2022 when the COVID-19 pandemic lasted (*p* < 0.0001). A similar trend was observed for PTC, when the incidence significantly increased to 13.99% in 2015 and then decreased to 1.38% in 2022 (*p* < 0.0001). At that time, the NIFTP category was introduced, and observation of PTMC began. The prevalence of FTC and MTC also increased until 2015 and then decreased. Significant differences in age, types of surgery, necessity of reoperation, and pTNM between PTCs and other types of TCs were observed. The average age was significantly lower in PTC patients than in patients with the remaining types of TC (*p* < 0.0001). Four milestones, including NIFTP introduction, the possibility of PTMC AS, personalized cancer medicine, and the COVID-19 pandemic, may have influenced the general statistics of TC.

## 1. Introduction

The rapid increase in the detection rate of thyroid cancer (TC) in recent years has caused many changes in the management of these malignancies. TC accounts for 3.4% of all cancers diagnosed annually worldwide [1]. Some data suggest that TC is increasing globally much faster than other malignant lesions [2,3]. This might be a consequence of the widespread use of ultrasonography (US) examination and ultrasound guided fine needle aspiration biopsy (UG-FNAB) procedures. Generally, this observation led to a globally highlighted discussion about the causes of such a situation. Some authors say that the higher incidence of TC is due to overdiagnosis [3]. Others believe that various additional factors may play a role in this observation, such as obesity and an increased exposure to ionizing radiation [4]. However, the fast increase in the occurrence of TC was observed to evolve to a point of slowing down and even reversing, which might be due to certain worldwide events. Some of these events might be the introduction of noninvasive follicular neoplasm with papillary-like nuclear features (NIFTP) as a TC type, the possibility of papillary thyroid microcarcinoma (PTMC) active surveillance (AS), the occurrence of personalized medicine in TC management, and, finally, the COVID-19 pandemic time. This phenomenon of TC occurrence is still debated and worthy of further investigation. When we start from TC with the most favorable prognosis, we have papillary TC (PTC) and follicular TC (FTC). Both types are known as well-differentiated TC (WDTC). Next, we have medullary TC (MTC), which is the only type that arises from C cells of the thyroid gland, poorly differentiated TC (PDTC), and anaplastic TC (ATC). ATC is the most aggressive and unfavorable form [1,2,3]. These five main forms of TC present tremendous variability in clinical course, histopathology, biological presentation, and, of course, prognosis. Four of these forms arise from thyroid follicular cells (PTC, FTC, PDTC, ATC). It is obvious that the heterogeneity of these four types of TC comes not only from histological diversity but also from genetic and epigenetic alterations, interactions between tumor cells and surrounding tissues forming specific microenvironments, and interhuman differences [1]. Even in ultrasound scans, many PTCs present irregular borders and infiltration of the surrounding thyroid physiological tissues. However, as a tumor assigned to the WDTC category, PTC very often contains a capsule, which makes it similar to noninvasive follicular thyroid neoplasm with papillary-like nuclear features (NIFTP). The introduction of NIFTP to thyroid tumor types might have had some influence on TC occurrence and surgical management.

Generally, WDTCs are indolent tumors characterized by an excellent prognosis. Almost all of them present a good response to standard therapeutic treatment. The 10-year survival rate is estimated to be 90% [2,3]. However, unfortunately, 10% of all patients with WDTC die because of this tumor [1].

The rapidly increasing number of TCs observed worldwide has forced scientists to debate whether it is a TC epidemic time or whether it is caused by the availability of better, more accessible diagnostic tools [2]. Moreover, the high diagnostic sensitivity of thyroid nodule stratification and its important role in the identification, management, and monitoring of these lesions was not limiting. This observation brought clinicians and scientists to the introduction of personalized cancer medicine in clinical practice. Although the majority of TCs are derived from one type of cell, these malignancies represent a very heterogeneous group of tumors.

The next important event, which might influence TC management, was the possibility of TC AS in clinical practice. Currently, AS might be a potential option for the management of patients with PTMC. Nevertheless, the decision to treat patients in this way should be made with information that reliably indicates that they have a low-risk tumor.

Because of these four important events, i.e., NIFTP introduction, possibility of PTMC AS, appearance of personalized medicine in clinical practice, and, finally, COVID-19 pandemic time, we decided to analyze the changes in the occurrence and management of TC during the past 14 years.

## 2. Materials and Methods

We retrospectively analyzed 5806 medical records of patients admitted and surgically treated in the Department and Clinic of General, Minimally Invasive and Endocrine Surgery due to thyroid nodules (TNs) between January 2008 and December 2022. The selection of the study group is presented as a flow diagram (Figure 1).

The study protocol was approved by the Institutional Review Board and Ethics Committee of Wroclaw Medical University, Wroclaw, Poland (No: KB 783/2023). All of our patients provided admission informed consent, which stipulated that the results may be used for research purposes. The data were analyzed retrospectively and anonymously from established medical records. The authors did not have access to identifying patient information or direct access to the study participants.

From a retrospective series of the thyroid surgical specimens archived in one center, 678 (11.7%) malignant thyroid tumors were obtained. All of the patients with malignant thyroid tumor resection had a minimum of one UG-FNAB performed before surgery. All cytological specimens were evaluated according to The Bethesda System for Reporting Thyroid Cytopathology (TBSRTC) introduced in 2009 [5] and reclassified in 2017 [6,7]. In all of the neoplasms, pTNM parameters were analyzed. All of the patients were evaluated, and the following parameters were recorded: sex, age, tumor size, tumor shape, echogenicity, microcalcifications, vascularity, and type of tumor (solitary, multifocal, bilateral).

We drew attention to some events, such as NIFTP introduction to thyroid tumor types, the possibility of PTMC observation in TC management, personalized medicine dissemination in recent years, and the COVID-19 pandemic. We checked whether “cancer screening activity” was depressed by the mentioned world events. We analyzed whether these events may have influenced the diagnostics, approach, and management of TC. Because of the opinion that all changes in the field of TC have been made mostly because of PTC, we also compared the most common type of TC, i.e., PTC, to the remaining types of TC in terms of incidence, clinical and pathological characteristics, and treatment.

### Statistical Analysis

Data analysis was conducted using MS Excel and Statistica 13.3 software (Tibco Software Inc., Palo Alto, CA, USA). The Kolmogorov-Smirnov test was employed to evaluate the data distribution. Descriptive data are presented as the number of observations (percent) or mean ± standard deviation (mean ± SD). The Chi-square test or Fisher’s exact test was used to analyze qualitative data. The Student’s *t*-test was employed to compare quantitative data between two independent study groups. All values of *p* < 0.05 were considered statistically significant.

## 3. Results

Overall, 678 out of 5806 (11.7%) patients were recruited as malignant tumors due to satisfying all required diagnostic criteria. Therefore, this study consisted of a group of PTC, FTC, MTC, undifferentiated TC (UTC), sarcoma, lymphoma, squamous cell carcinoma, myeloma, and secondary tumors in a 14-year-long period (2008–2022). Table 1 presents the demographics and clinical and tumor characteristics of the TC group and two subgroups of TC: PTC and the remaining types of TC (Table 1).

There were 581 (85.69%) female patients and 97 (14.31%) males in the TC group. The average age at diagnosis was 51.66 ± 15.98 years old (range 18–81 years). The largest subgroups of TC patients were PTC—579 patients (85.39%), FTC—31 (4.57%), and MTC—24 (3.53%). Significant differences in age, type of surgery, necessity of reoperation, pTNM, pT, pN, and pM were observed between PTC and patients with other types of TC (Table 1). The average age was significantly lower in PTC patients than in patients with the remaining types of TC (*p* < 0.0001). Total thyroid resection was performed significantly more often in PTC patients than in patients with other types of TC (*p* < 0.0001). In the majority of PTC cases, we were able to perform radical surgeries, whereas in the other types of TC, like medullary, undifferentiated, sarcomas, squamous cell carcinomas, or even follicular TCs, often we were not. In many of these last cases, reoperations had to be performed. Patients with other types of TC significantly more often required reoperation due to nonradical primary procedure (*p* = 0.002). In the PTC subgroup, significantly higher rates of pTNM stage I were observed, and in other types of TC, the prevalence of pTNM stage IV was significantly higher than that in PTC (*p* < 0.0001). Among the subgroup of patients with other types of TC, the rates of pT4b, pN1b, and pM1 were significantly higher in comparison to PTC patients (*p* < 0.0001 for all).

The UG-FNAB results were available in all cases (100%). The selected ultrasound features in the total group of TC, PTC, and other types of cancer are presented in Table 2. A significantly higher rate of tumor size above 5 mm, irregular tumor shape, hypoechogenicity, presence of microcalcifications, and high vascularity was observed in the subgroup of patients with other types of TC than in patients with PTC (*p* < 0.05 for all).

The prevalence rates of benign thyroid tumors and thyroid cancers (TCs) from 2008 to 2022 are shown in Table 3 and Figure 2. The prevalence of TC significantly increased from 5.15% in 2008 to 13.84% in 2015, and then the incidence of TC significantly decreased to 1.33% in 2021 and 2022 (*p* < 0.0001).

A similar trend was observed for PTC, where the incidence of cases significantly increased to 13.99% in 2015 and then significantly decreased to 1.38% in 2021–2022 (*p* < 0.0001) (Table 4 and Figure 2). The prevalence of FTC and MTC also increased until 2015, before a decrease in the rate of cases of these types of TC was observed.

In Figure 3, we present the incidence of benign thyroid tumors with respect to the sex of patients in 2008–2022 (Figure 3).

In Figure 4, we present the incidence of malignant thyroid tumors, i.e., all TC patients and PTC patients with respect to the sex of individuals in 2008–2022 (Figure 4).

In Figure 5 and Figure 6, we present the incidence of malignant thyroid tumors, i.e., all TC patients and PTC patients with respect to age (<55 years old vs. >55 years old) and the incidence of PTC patients with respect to tumor size (<5 mm vs. >5 mm) in 2008–2022 (Figure 5 and Figure 6, respectively).

Finally, one of the most interesting observations is presented in Figure 7. We show the incidence of PTC patients and remaining TC patients regarding pTNM classification in 2008–2022 (Figure 7).

## 4. Discussion

In recent years, the number of TCs has increased; however, almost 60–70% are low-risk PTMCs [8,9,10,11]. However, heterogeneity, even in such a homogenous group of tumors, can also be observed [9,10,11]. First, there might be multiple lesions. Multifocal unilateral or bilateral PTC is also not very rare [12]. This feature of PTC is not always connected with heterogeneity at the molecular and gene levels; however, some authors confirm the high rate of molecular variation within one tumor [13,14,15]. In our study, the majority of TCs in recent years were PTCs below 1 cm in dimension (PTMC, pT1a). The next largest group of TCs was also PTC, but in lesion sizes of above 1 cm to 2 cm in diameter (pT1b). We did not observe it in the remaining TC types. In this group of patients, the smallest tumors (pT1a) constituted the smallest group. Therefore, we confirmed in our analysis that in recent years, the majority of TCs are still small tumors, mostly less than 1 cm in diameter.

As mentioned previously, WDTCs are generally indolent tumors characterized by an excellent prognosis, although some authors observe a high rate of WDTC recurrence worldwide [16,17]. A completely different situation occurs in the case of PDTC and ATC. Both of these tumors present a dramatically poor prognosis. One of the reasons for this situation is resistance to radioiodine therapy. In our analysis, among PTC patients, the largest group comprised patients who did not require surgery after the first surgical intervention. This might be because of the previously mentioned characteristic, that many of the patients were diagnosed in the very low stage of disease. Unfortunately, we did not observe such a situation in the case of other TCs. In this group of neoplasms, many individuals required reoperation due to nonradical primary treatment or tumor recurrence.

The majority of PTC cases are associated with additional TC foci localized in the same or second lobe of the thyroid gland. This characteristic pathological feature of PTC is estimated for 18–87% of all PTCs [1,18,19]. For many years, debate might be observed regarding whether such PTC features come from autonomous multiple independent tumors or are a consequence of intrathyroidal spread of the primary PTC focus. Currently, on the basis of many studies, both theories of PTC multiplicity are still valid [20,21]. However, some authors emphasize that while the BRAF V600E mutation is observed in the primary tumor, it is very often not observed in LNM and coexisting lesions [22]. In our opinion, there might also be a situation in which the ipsilateral or contralateral PTC additional foci without the BRAF V600E mutation metastasize to the lymph nodes, so they are not presented in the cancer genetic profile. However, we can also suggest that this mutation can develop de novo in metastatic PTC cells. We estimate that the multifocality of neoplasms is not characteristic of PTC only. We observe the same characteristic in the remaining TCs. A different situation exists regarding bilaterality. This histopathological feature remains characteristic of PTC. In 2022, the WHO introduced the 5th edition of the Classification of Endocrine and Neuroendocrine Tumors, which is related to the thyroid gland [23]. One of the most important and newest classification change is the division of thyroid tumors for BRAF-like malignancies represented by PTC, with many morphological subtypes and RAS-like malignancies represented by an invasive encapsulated follicular variant of PTC and FTC [23].

A rapidly increasing number of TCs has been observed worldwide [2,16,24]. Currently, we notice a widespread application of neck ultrasound examination and, consequently, a large number of UG-FNABs of thyroid nodules [25]. This observation occurs despite the proposal and introduction of the American Thyroid Association (ATA) Guidelines in 2016 [26,27]. In one of our previous studies, we called this phenomenon “cancer screening activity” [25]. Indeed, in our analysis, we observed this phenomenon in our study population up to 2016. At that point, new cases, especially small PTCs, were diagnosed. After that year, some new world events occurred. They probably influenced this phenomenon and changed the TC number tendency.

Currently, the treatment of TC patients depends on the characteristics of histopathological samples obtained from tumors after surgery. Such a clinical approach applies to all forms of TC, especially follicular types, and all forms of WDTCs. AS is one of the potential options for the management of patients with PTMC. Generally, PTMC grows slowly and has indolent biological behavior. The rate of local recurrence is estimated as 2–6%, and distant metastasis is extremely rare and is observed in 1–2% of cases [28,29]. Some authors suggest that even when local or regional metastases and disease progression appear, such a clinical situation does not worsen the prognosis [30]. It is obvious that delayed surgery is mandatory. On the basis of many studies, AS was introduced in some countries, such as the United States, Korea and Japan, as a potential management strategy for low-risk PTMC and as an alternative to immediate surgery [26,31,32,33,34,35]. However, in many countries, immediate surgery is the first-line and routine treatment in many cases of PTMC [26,36]. The authors highlight some advantages of this option, such as the simple removal of the malignant tumor, feasible and easier follow-up, decreased risk of secondary surgeries, and, perhaps the most valuable for many patients, elimination of the inherent anxiety connected with awareness of having a malignant tumor. On the other hand, we must remember some potential complications that might occur after surgical treatment [37,38,39]. There is an open question or unsolved dilemma regarding which quality of life is lower: living with the potential complications after surgery or living with an awareness of malignant tumors inside. However, the main question stands—not about tumor progression or local or regional metastasis, because it has been estimated that delayed surgery guarantees favorable outcomes; rather, the main question relates to the risk of potential distant metastasis in the AS period and whether the prognosis in such a clinical situation is worse. Some autopsy studies have revealed the high prevalence of small, subclinical TC, particularly PTMC [40]. Hugen et al. [41] showed in a national Dutch autopsy study that distant metastases were revealed in 8.9% of cases with PTC and in 19% of individuals with FTC in whom TC was first diagnosed on autopsy. In our analysis, distant metastases were observed much less frequently in the PTC group than in the remaining TC patients (3.45% vs. 26.53%). Therefore, this observation confirms that AS might be a good therapeutic option for some PTMC patients. In Poland, the discussion about AS as a potential therapeutic option for some individuals harboring PTMC started in 2016 [25]. Therefore, after this point in time, a slight decrease in the number of new TC cases was observed. Thyroid gland US examination enables the detection of even small TCs. At the time of this diagnosis, the main question appears whether to observe or excise the tumor. This is the crucial question addressed by surgeons at this time in PTMC management. Ito and Tuttle stated that AS is a safe alternative to immediate surgery, but only for selected patients [42,43]. Therefore, the question is which patients? How do we select the patients in routine clinical practice and possibly do so without increasing costs? The second question is whether patient selection is effective. Many younger individuals prefer AS over immediate surgery; however, according to some authors, younger age is associated with a higher rate of TC growth [30,43]. Heterogeneity of thyroid cells applies to tumor and stromal tissues. Tumor cells may exhibit follicular differentiation, perifollicular differentiation, or both types of differentiation or dedifferentiation. Due to the two main hypotheses of carcinogenesis, we can form some clinically pragmatic observations. The first is that tumor growth is a consequence of genome instability of somatic cells, which may lead to the selection of more aggressive clones and thus more aggressive TC types [44,45]. This is known as a multistep carcinogenesis model [1,44,45]. The second hypothesis is that in the primary tumor, there are small populations of stem cells, which, after certain mutations or genetic and epigenetic transformations, may produce phenotypically totally different cells of TC, very often more aggressive ones. This second theory, known as the cancer stem cell model, was first presented by Mitsutake et al. [46] in 2007. On the basis of these two theories, we can assume that PDTC and ATC may come from WDTCs and, consequently, when we treat patients with WDTC, some of them might be patients with early-stage PDTC or ATC. We are aware that this is a controversial question; however, in our clinical practice, there is some evidence to draw attention to this issue. PDTC is characterized by expansive growth, an incomplete capsule, an increased number of tumor-associated macrophages, and low lymphocyte and dendritic cell infiltration compared to PTC [47,48]. Additionally, the BRAF V600E mutation is also frequently observed in PDTC connected with PTC nuclear features or even PTC foci. Currently, ATC is considered the most heterogeneous TC. Moreover, it is still an extremely invasive thyroid neoplasm with extensive infiltration of the lymph nodes and surrounding soft neck tissues and organs. One “optimistic” feature, if we can express it in this way, is that similarly to encapsulated PDTC, encapsulated types of ATC sometimes might present a slightly better prognosis with an overall survival time up to 57 months [49,50]. In our group of ATC patients, we noticed that a well-defined, nonruptured ATC capsule was a good prognostic factor of ATC regardless of some other ultrasound and pathologic features.

In 2017, the WHO established a new classification for endocrine tumors [51,52,53]. It elaborated and published the criteria for a new endocrine tumor, which was named noninvasive follicular thyroid neoplasm with papillary-like nuclear features (NIFTP). Unfortunately, NIFTP can be described only on the basis of the surgical specimens, and its final histological diagnosis is possible after surgery. The great utility and usefulness of this new classification, especially in clinical practice, is well established. In our study, according to the ATA classification, after 2017, we excluded all NIFTP entities from the TC group and included them in the benign tumor group. This might be the next reason for the decrease in TC number after 2016. As clinicians, we should know that NIFTP has to exhibit a well-defined capsule, a sharply separated nodule mass from the thyroid tissue, and no infiltration. The pattern of architecture of this nodule is follicular but with nuclear characteristics, which are similar to those of classical PTC. Despite a precise description of NIFTP and its great clinical utility, there are some fundamental questions that still do not have accurate answers. For example, the minimal and maximal sizes of NIFTPs are not specified; however, many scholars describe these tumors as lesions with dimensions of 1 cm or more [54,55,56]. Interestingly, NIFTP is currently recognized as a benign tumor; however, it is still very often treated as TC. Generally, these tumors present an indolent course and do not present a genetic profile characteristic of malignant entities. Therefore, in our patients, all of those who harbored NIFTP underwent molecular testing according to BRAF 600 mutation to exclude TC diagnosis. This test is performed in an oncological center, in which all patients with NIFTP are reconsulted and potentially treated.

In the analyzed period, we noticed a fast increase in TC incidence up to 2015–2016. We observed that the occurrence slowed down and even reversed after this point in time—especially in small entities of PTC, in likely response to some clinical, histopathological, sociological, and epidemiological events. At the very beginning of our observation period, high-resolution imaging techniques were introduced to detect even small TNs. Because of widespread US and UG-FNAB, an increased detection rate of PTC was observed. However, it was not connected with any benefits, such as reduced mortality risk [57]. Given such an observation, ATA guidelines did not recommend a biopsy of small nodules, especially below 5.0 mm in diameter [26]. However, in our patients, we did not observe important differences after this time in the PTC group or in the remaining TCs.

Despite the indolent nature of some PTCs and the majority of PTMCs, the words “cancer” or “carcinoma” included in their names cause unnecessary fear and anxiety in patients. Moreover, these terms very often force surgeons to perform aggressive procedures such as total thyroidectomy with lymphadenectomy of the central and even lateral compartments. However, currently, the treatment of some benign tumors, such as NIFTP, is still controversial. Some authors recommend total thyroidectomy in cases of NIFTP because, in their opinion, this tumor is often multifocal or bilateral with lymph node metastases [58]. Nevertheless, others say that PTC and NIFTP, with diameters of 1–4 cm, surgically treated by thyroidectomy or lobectomy with lymphadenectomy would provide comparable oncologic outcomes [59]. In cases of larger tumors, i.e., more than 4 cm in diameter, the main question is whether only surgical treatment is sufficient or whether complete radioiodine therapy with TSH suppression should also be added. ATA guidelines recommend two-step therapy options, i.e., surgery with radioiodine therapy completion [26]. However, there are also authors who do not accept this approach and say that even larger tumors may be treated less aggressively [60]. Generally, the majority of NIFTP tumors, even those with dimensions greater than 4 cm, are treated as indolent nodules, and radioiodine therapy or suppressive doses of levothyroxine are avoided. In our group of patients with NIFTP diagnosis, surgery was provided for everyone as the only target therapy. During the observation time, we did not observe any recurrence or metastases. However, our observation time may be too short to form any conclusions. In any event, our observations are in accordance with some other authors, who also followed up patients with multifocal or bilateral NIFTP [61]. Others emphasize that because of these clinical and pathological characteristics of NIFTP, patients should be strictly observed after lobectomy [62]. We absolutely agree with these authors, and we practice according to these recommendations.

PTC very often exists as an accompanying lesion, i.e., one tumor diagnosed presurgically, as PTC is found microscopically after surgery to be composed of a minimum of two subtypes of PTC. Even homogenous types, such as the classical variant of PTC (CVPTC), are very often accompanied by the follicular variant of PTC (FVPTC) [1]. It is extremely important to diagnose all of the components of PTC nodules if one of them represents a more aggressive type. Some authors emphasize that such uneventful characteristics, such as metastasis or recurrence, are just produced by its aggressive part [63]. LiVolsi [63] noticed that a worse prognosis of PTC is made by the higher percentage of coexistence of tall-cell variant PTC (TCVPTC). However, as some authors say, it is not established what minimal percentage of the component of TCVPTC is required to diagnose the cancer as TCVPTC, and they say approximately 10–70% of cells are reported [64]. According to the latest data, a minimum of 50% of TCVPTC in PTC nodules should be diagnosed as TCVPTC [63]. Of course, there might be some tumors that include two or more subtypes or variants of PTC, and, more interestingly, both might be on the same level of clinical importance. Baloch et al. [65] described a patient with a malignant tumor consisting of TCVPTC and Hurthle cell cancer. It had clinical consequences because TCVPTC metastasized to regional lymph nodes, but Hurthle cell cancer metastases were observed in the lungs [65]. Several years ago, we performed studies in which we analyzed many cases of PTC and PTMC with completely different clinical courses and prognoses [15,66]. However, we did not have accurate and detailed histopathological descriptions of every single microcarcinoma. Currently, it is well established that for different clinical courses, potential metastasis and recurrence depend on histological subtypes and variants. In our group of patients with a very unfavorable clinical course, after some specimen reclassification, we diagnosed three cases of hobnail variant PTC (HVPTC) in heterogeneous PTC cases along with TCVPTC and anaplastic thyroid cancer. Baloch et al. [67] assessed that HVPTC may account for approximately 30% of all PTC subtypes. Moreover, approximately 50% of HVPTC lesions represent distant metastases to the brain, lungs, liver, and bones [67]. Other authors proved that HVPTC was significantly associated with lymph node metastasis (LNM) [68].

It was estimated that appropriate individuals for AS are patients with characteristics as follows: well-defined solitary nodules equal to or less than 1–1.5 cm in dimension, diagnosed with PTC with at least 2 mm of noninfiltrated thyroid tissue surrounding the tumor, without pathological lymph nodes, and older age [69,70]. The molecular profile of TC is currently not mandatory and is considered as additional information in AS and patient selection [71]. Tuttle et al. [69] estimated that patients with PTC and multiple driver events, such as BRAF V600E + TERT and RAS + TERT, were not appropriate candidates for AS. Regarding some clinical and ultrasound features, Sakai et al. [72] noticed that high vascularity and calcifications predispose tumors to faster growth, which makes them inappropriate for AS.

In our opinion, AS is one of the most responsible methods of PTC patient management. It requires many ultrasound examination skills on the part of the attending physicians. The patient has to agree to long-term observation methods, so a high level of understanding of PTC behavior is absolutely needed. Such a clinical approach to PTC patients is called personalized medicine, which has been widely promoted in Poland since 2017 [25].

The next clinical dilemma connected with patients with small PTC, which is currently also under debate, is the extent of surgery. As far as recommendations, for small PTC, i.e., equal to or below 1.0 cm in diameter, guidelines are clearly established. Larger tumors are treated more variably. During the past 14 years of observation, the approach to diagnosis and treatment of patients with thyroid cancer has changed based on better diagnosis and faster detection. Although thyroid surgery, radioactive iodine therapy, and TSH suppression continue to be the mainstay of treatment, current knowledge has allowed the treatment of people with low-risk differentiated thyroid cancer to be de-escalated. Treatment options for patients with aggressive thyroid tumors have also been expanded. In addition, over the past 14 years, advances in knowledge of the molecular aspects of thyroid cancer have improved the diagnosis of TC and enabled individualized treatment options for selected patients with the most aggressive form of the disease [73]. The guidelines of many societies around the world reflect these changes and include a focus on adopting a more individualized approach to clinical management. Some management aspects of TC treatment, which were impossible some years ago, such as AS of selected small TCs or monitoring small TNs without UG-FNAB, are currently implemented and are even clinical routine in some departments. Regarding the surgical treatment of TC, there is more acceptance of hemithyroidectomy for low-risk TC. However, we must remember the challenges resulting from these new approaches to TC management, such as long-term follow-up costs, patient and clinician anxiety, and uncertainty in thyroglobulin (Tg) monitoring after nonradical treatment. However, personalized medicine in the management of TC is currently well established, and it will probably remain for a long time.

Our study has some limitations. First, it is a retrospective study, so some inaccuracies typical of such studies were unfortunately present. Second, this work was performed at a single institution. Third, the number of patients was not very high. Fourth, one of the inclusion criteria of this study was obtaining histopathology results, so the study included selection bias because we evaluated only patients with malignant tumors who underwent surgery. However, the histopathology results of all patients were mandatory for this study to form any conclusions. Fifth, the analyzed patients did not undergo molecular tests, so no correlation to pathological diagnosis was estimated. We are aware that this information would be the most valuable, especially in patients in whom AS and personalized medicine approaches were taken into consideration.

## 5. Conclusions

In conclusion, during the time when “cancer screening activity” was widely observed, four world events, i.e., the possibility of PTMC AS in clinical practice, NIFTP introduction to TNs and exclusion from TC types, personalized medicine occurrence in the field of TC approach, and COVID-19 pandemic time, were observed. They may have influenced the general statistics of TC, including the number of new cases and trends in clinical management.

## Figures and Tables

**Figure 1 cancers-15-04941-f001:**
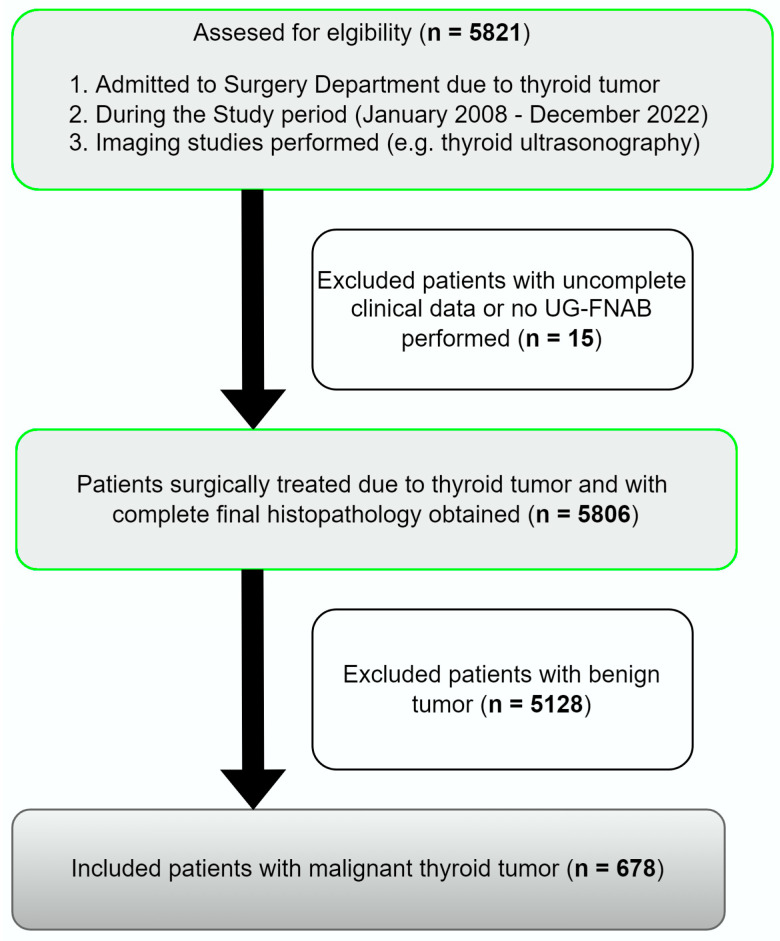
Selection of the study group from 5821 individuals referred for surgery from 2008 to 2022. All participants underwent a minimum of one UG-FNAB. All evaluated patients underwent surgery, and histopathology results were obtained in all cases. A total of 678 patients with malignant tumors were included and analyzed. UG-FNAB: ultrasound guided fine needle aspiration biopsy.

**Figure 2 cancers-15-04941-f002:**
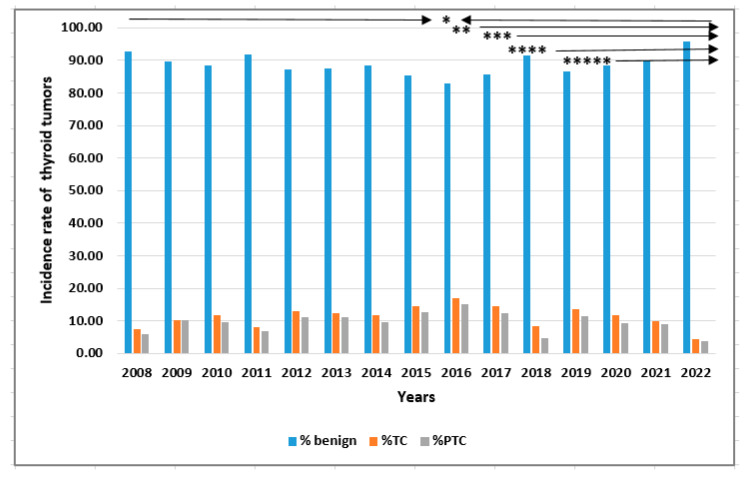
The incidence rate of thyroid tumors benign, TC, and PTC in 2008–2022. Rates were calculated as number of cases per number of all patients in study year. * apogee of “cancer screening activity” time, ** possibility of PTMC AS in clinical practice, *** NIFTP excluded from TC types, **** personalized medicine recommended, ***** COVID-19 pandemic time. PTMC AS: papillary thyroid microcarcinoma active surveillance.

**Figure 3 cancers-15-04941-f003:**
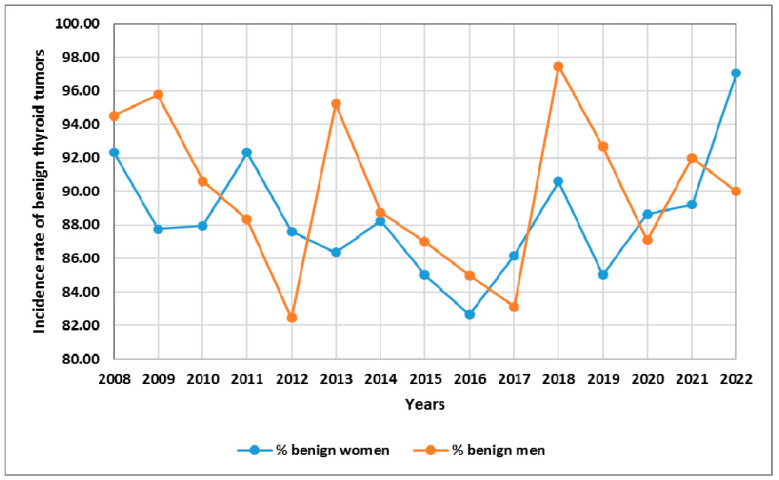
The incidence of benign thyroid tumors with respect to the sex of patients in 2008–2022. Rates were calculated as the number of cases per number of all patients in the study year.

**Figure 4 cancers-15-04941-f004:**
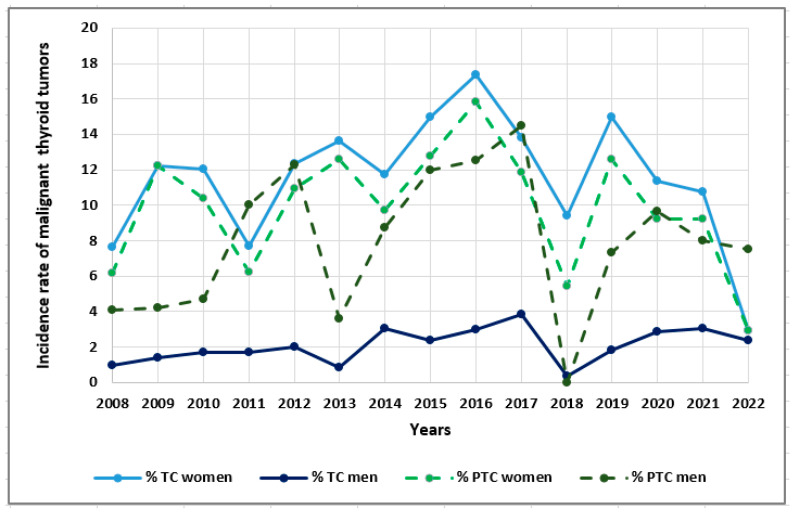
The incidence of malignant thyroid tumors TC and PTC with respect to the sex of patients in 2008–2022. Rates were calculated as the number of cases per number of all patients in the study year.

**Figure 5 cancers-15-04941-f005:**
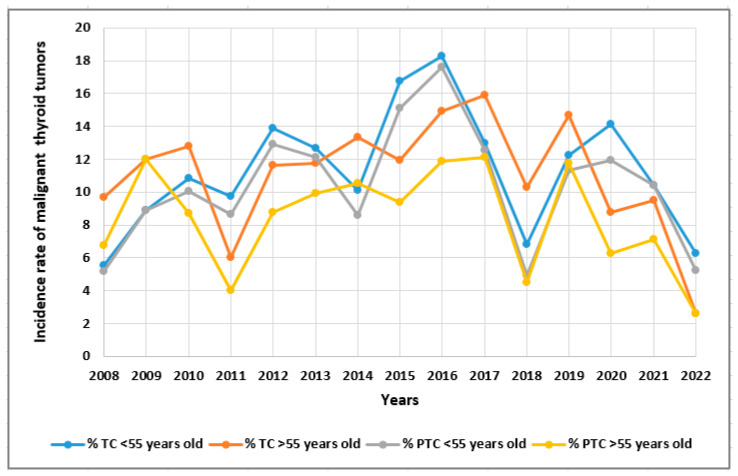
The incidence of malignant thyroid tumors TC and PTC with respect to age (<55 years old vs. >55 years old) in 2008–2022. Rates were calculated as the number of cases per number of all patients in the study year.

**Figure 6 cancers-15-04941-f006:**
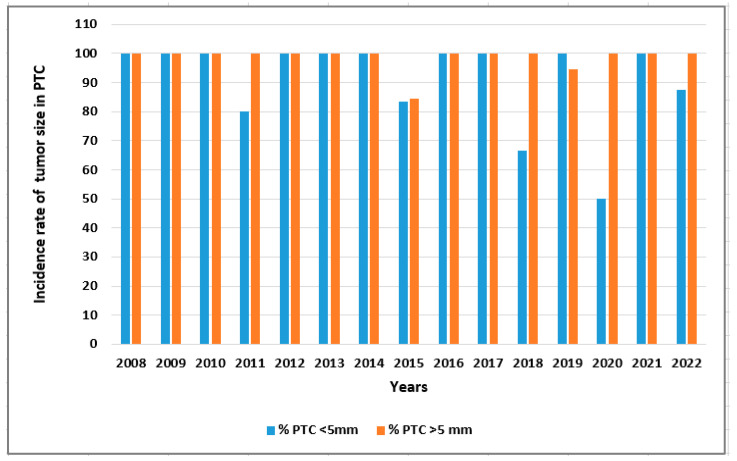
The incidence of PTC with respect to tumor size (<5 mm vs. >5 mm) in 2008–2022. Rates were calculated as the number of PCT cases per number of TC patients in the study year.

**Figure 7 cancers-15-04941-f007:**
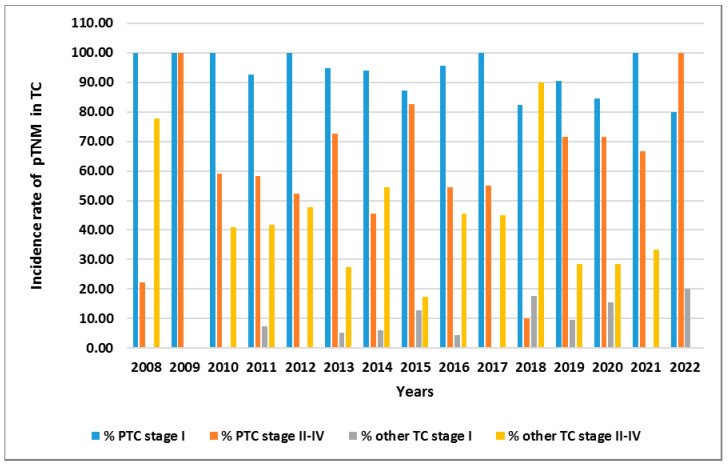
The incidence of PTC and other TC with respect to pTNM (stage I vs. stage II–IV) in 2008–2022. Rates were calculated as the number of PCT cases/other TC cases per number of TC patients in the study year.

**Table 1 cancers-15-04941-t001:** Demographic, clinical, and tumor characteristics of all patients with TC, papillary thyroid cancer (PTC), and other types of cancer.

Parameters	Total TC Patients (*n* = 678)	PTC Patients (*n* = 579)	Other Types of TC (*n* = 99)	*p* Value
	N (%) or Mean + SD	N (%) or Mean + SD	N (%) or Mean + SD	
Sex:				0.133
Female	581 (85.69)	501 (86.53)	80 (80.81)	
Male	97 (14.31)	78 (13.47)	19 (19.19)	
Age (years)	51.66 + 15.98	50.25 + 15.20	59.92 + 17.95	<0.0001 *
Age:				<0.0001 *
<55 years old	385 (56.78)	355 (61.31)	30 (30.30)	
>55 years old	293 (43.22)	224 (38.69)	69 (69.70)	
Type of surgery:				<0.0001 *
Total	474 (69.91)	426 (73.58)	48 (48.48)	
No total	204 (30.09)	153 (26.42)	51 (51.52)	
Reoperation needed:				0.002 *
No	502 (74.04)	441 (76.17)	61 (61.62)	
Yes	176 (25.95)	138 (23.83)	38 (38.38)	
Histological type of cancer:		579 (100.00)		-
Papillary (PTC)	579 (85.39)		-	
Follicular (FTC)	31 (4.57)		31 (31.31)	
Medullary (MTC)	24 (3.53)		24 (24.24)	
Undifferentiated	14 (2.06)		14 (14.14)	
Sarcoma	3 (0.44)		3 (3.03)	
Secondary	10 (1.47)		10 (10.10)	
Lymphoma	12 (1.76)		12 (12.12)	
Squamous cell carcinoma	4 (0.58)		4 (4.04)	
Myeloma	1 (0.14)		1 (1.01)	
pTNM:				<0.0001 *
I	501 (73.89)	473 (81.69)	27 (27.55)	
II	90 (13.27)	74 (12.78)	16 (16.33)	
III	42 (6.19)	24 (4.15)	18 (18.37)	
IV	45 (6.63)	8 (1.38)	37 (37.76)	
pT:				<0.0001 *
pT1a	256 (37.75)	245 (42.76)	9 (9.09)	
pT1b	276 (40.70)	245 (42.76)	29 (29.29)	
pT2	78 (11.50)	61 (10.65)	15 (15.15)	
pT3	24 (3.54)	13 (2.27)	11 (11.11)	
pT4a	16 (2.36)	3 (0.52)	13 (13.13)	
pT4b	26 (3.83)	5 (0.87)	21 (21.21)	
pTm	2 (0.29)	1 (0.17)	1 (1.01)	
pN:				<0.0001 *
pN0	427 (62.97)	386 (66.67)	41 (41.41)	
pN1a	184 (27.14)	158 (27.29)	26 (26.26)	
pN1b	35 (5.16)	13 (2.25)	22 (22.22)	
pNx	32 (4.72)	22 (3.80)	10 (10.10)	
pM:				<0.0001 *
pM0	568 (83.78)	506 (87.39)	61 (62.24)	
pM1	46 (6.78)	20 (3.45)	26 (26.53)	
pMx	64 (9.43)	53 (9.15)	11 (11.22)	

TC: thyroid cancer, PTC: papillary thyroid cancer, FTC: follicular thyroid cancer, MTC: medullary thyroid cancer, *: statistically significant.

**Table 2 cancers-15-04941-t002:** Selected ultrasound features of all TCs, papillary thyroid cancer (PTC), and other types of cancer.

Parameters	Total TC Patients (*n* = 678)	PTC Patients (*n* = 579)	Other Types of TC (*n* = 99)	*p* Value
	N (%)	N (%)	N (%)	
Tumor size:				0.049 *
<5 mm	294 (43.36)	225 (38.89)	69 (70.00)	
>5 mm	384 (56.64)	354 (61.11)	30 (30.00)	
Tumor shape:				<0.0001 *
Regular	294 (43.36)	284 (49.13)	10 (10.20)	
Irregular	384 (56.64)	295 (50.87)	89 (89.80)	
Echogenicity:				0.0001 *
Hyperechoic	120 (17.69)	116 (20.03)	4 (4.08)	
Hypoechoic	558 (82.15)	463 (79.97)	95 (95.92)	
Microcalcifications:				<0.0001 *
No	275 (40.56)	259 (44.73)	16 (16.16)	
Yes	403 (59.44)	320 (55.27)	83 (83.84)	
Vascularity:				<0.0001 *
Low	307 (45.28)	295 (51.04)	12 (12.24)	
High	371 (54.72)	283 (48.96)	87 (87.76)	
Type of tumor:				0.132
Solitary	488 (71.98)	423 (73.01)	65 (65.66)	
Multifocal	190 (28.02)	156 (26.99)	34 (34.34)	
Bilateral:				0.060
No	626 (92.33)	529 (91.52)	96 (96.97)	
Yes	52 (7.67)	49 (8.48)	3 (3.03)	

TC: thyroid cancer, PTC: papillary thyroid cancer, *: statistically significant.

**Table 3 cancers-15-04941-t003:** The prevalence of benign thyroid tumors and thyroid cancers (TCs) in 2008–2022. Descriptive data are presented as the number of observations (percent).

Year	Benign Tumors	Thyroid Cancers	All Patients
2008	443 (8.64)	35 (5.15)	478 (8.23)
2009	342 (6.67)	39 (5.74)	381 (6.56)
2010	372 (7.26)	49 (7.22)	421 (7.25)
2011	438 (8.54)	39 (5.74)	477 (8.22)
2012	479 (9.34)	71 (10.46)	549 (9.46)
2013	491 (9.58)	69 (10.16)	560 (9.65)
2014	334 (6.50)	44 (6.63)	378 (6.51)
2015	547 (10.67)	94 (13.84)	641 (11.04)
2016	397 (7.74)	81 (11.93)	478 (8.23)
2017	381 (7.43)	64 (9.43)	445 (7.67)
2018	290 (5.66)	27 (3.98)	317 (5.46)
2019	180 (3.51)	28 (4.12)	208 (3.58)
2020	152 (2.96)	20 (2.95)	172 (2.96)
2021	81 (1.58)	9 (1.33)	90 (1.55)
2022	201 (3.92)	9 (1.33)	210 (3.62)
N (%) for groups	5128 (100.00)	678 (100.00)	5806 (100.00)
N (%) for total	5128 (88.32)	678 (11.68)	5806 (100.00)

**Table 4 cancers-15-04941-t004:** The prevalence of types of TC in 2008–2022. Descriptive data are presented as the number of observations (percent).

Year	PTC	FTC	MTC	Undifferentiated	Sarcoma	Secondary TC	Lymphoma	SCC	Myeloma	All
2008	28 (4.84)	1 (3.23)	1 (4.17)	1 (7.14)	1 (33.33)	1 (10.00)	2 (16.67)	-	-	35 (5.16)
2009	39 (6.74	-	-	-	-	-	-	-	-	39 (5.75)
2010	40 (6.91)	2 (6.45)	2 (8.33)	4 (28.57)	-	-	1 (8.33)	-	-	49 (7.23)
2011	32 (5.53)	3 (9.68)	2 (8.33)	1 (7.14)	-	1 (10.00)	-	-	-	39 (5.75)
2012	61 (10.54)	4 (12.90)	1 (4.17)	1 (7.14)	1 (33.33)	2 (20.00)	1 (8.33)	-	-	71 (10.47)
2013	63 (10.88)	3 (9.68)	-	1 (7.14)	-	2 (20.00)	-	-	-	69 (10.18)
2014	36 (6.22)	2 (6.45)	2 (8.33)	-	1 (33.33)	1 (10.00)	2 (16.67)	-	-	44 (6.49)
2015	81 (13.99)	5 (16.13)	7 (29.17)	-	-	-	1 (8.33)	-	-	94 (13.86)
2016	73 (12.61)	-	3 (12.50)	2 (14.29)	-	3 (30.00)	-	-	-	81 (11.95)
2017	55 (9.50)	-	1 (4.17)	1 (7.14)	-	-	3 (25.00)	3 (75.00)	1 (100.00)	64 (9.44)
2018	15 (2.59)	6 (19.35)	2 (8.33)	2 (14.29)	-	-	1 (8.33)	1 (25.00)	-	27 (3.98)
2019	24 (4.15)	1 (3.23)	2 (8.33)	1 (7.14)	-	-	-	-	-	28 (4.13)
2020	16 (2.76)	4 (12.90)	-	-	-	-	-	-	-	20 (2.95)
2021	8 (1.38)	-	-	-	-	-	1 (8.33)	-	-	9 (1.33)
2022	8 (1.38)	-	1 (4.17)	-	-	-	0 (0.00)	-	-	9 (1.33)
N (%) for subgroups	579 (100.00)	31 (100.00)	24 (100.00)	14 (100.00)	3 (100.00)	10 (100.00)	12 (100.00)	4 (100.00)	1 (100.00)	678 (100.00)
N (%) for total	579 (85.40)	31 (4.57)	24 (3.54)	14 (2.06)	3 (0.44)	10 (1.47)	12 (1.77)	4 (0.59)	1 (0.15)	678 (100.00)

TC: thyroid cancer; PTC: papillary thyroid cancer; FTC: follicular thyroid cancer; MTC: medullary thyroid cancer; SCC: squamous cell carcinoma.

## Data Availability

The datasets used and/or analyzed during the current study are available from the corresponding author upon reasonable request.

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
