# Peer review of "The Incidence Trend and Management of Thyroid Cancer—What Has Changed in the Past Years: Own Experience and Literature Review"

_cancers, 2023, doi:10.3390/cancers15204941_

Round 1
Reviewer 1 Report
This is a large retrospective single –center study aimed at studying the dynamic of thyroid cancer diagnosis over more than 10 years.
Minor issues:
- I would suggest considerably shortening both the introduction and the discussion to include only most relevant information
2.Table 1 please align correctly histological type column 2 and 3 (it is not clear how many PTC vs follicular)
Significant questions
1.The percentages of tumor <5 mm in whom FNAB was performed preoperatively are extremely high. Please explain in detail the reasoning behind the indication for FNAB in all these patients and also, for the 38% with very small PTC the reason for proceeding to thyroidectomy
2. Did this practice of performing FNAB in very small lesions not change itself during the years of the study? This might significantluy alter the incidence results
3. The discussion revolves around the option of AS but the results do not reflect at all this, please clarify
4. “Total thyroid resection was performed significantly more often in PTC patients than in patients with other types of TC” –please provide an explanation
Author Response
Journal: Cancers September 30, 2023
MDPI
Special Issue: “Classification, Risk Assessment and Clinical Management of Malignant Nodules.”
Guest Editors: Prof. Claudio Spinelli, Dr Marco Ghionzoli
Dear Editors and Reviewers,
At the very beginning we would like to thank you very much for the possibility to re-submit our revised manuscript entitled “The Incidence Trend and Management of Thyroid Cancer – What has Changed in the Last Years: Own Experience and Literature Review.” for consideration as an original article in Special Issue “Classification, Risk Assessment and Clinical Management of Malignant Nodules.” Thank you very much for considering it for potential publication in Cancers.
We would like to thank you for the very thorough reviews and for the advices and constructive criticism, which have been valuable for improving our paper. All of the suggestions for changes and improvements were very helpful to us, and we have revised the manuscript according to the recommendations made in the reviews. All of the changed, deleted and added portions of the manuscript are marked by using Track Changes. According to the reviewers’ instructions we corrected our manuscript point-by-point as follows:
Reviewer 1
At the very beginning we would like to thank you very much for your opinion about “large (…) study.” Indeed, we observed our patients and followed them for more than 15 years. Thank you so much, that you noticed and appreciated it. Thank you.
Minor issues
Ad. 1 Thank you for this suggestion, we tried to shorten the manuscript as much as it possible. Thank you.
Ad. 2 Thank you very much for this remark. We aligned the column 2 and 3. It happened during the final re-edition of the tables in the manuscript. Thank you
Significant questions
Ad. 1 Dear Reviewer, from the point, when we started to observe our patients, we noticed in our population the phenomenon, which we called “cancer screening activity” and we described it in our article by Kaliszewski et al.. Cancer screening activity results in overdiagnosis and overtreatment of papillary thyroid cancer: A 10-year experience at a single institution. PLoS One 2020, 15, e0236257; DOI:10.1371/journal.pone.0236257. It was well received in USA and in other countries. We noticed, that the number of ultrasound examinations, which revealed many thyroid lesions, especially very small nodules, was very high. In our country, access to ultrasound examination is not dificult, so many individuals have it done. Thus, when clinicians consult the patients with very small thyroid tumors decsribed in ultrasonography as EU-TIRADS-PL 4 or 5, the next step established and undertaken by them is sending these patients to UG-FNAB. So, it is the reason, why so many patients with PTC had the malignant tumor below 5mm in diameter. We noticed it ealier, and decided to analyse it. As far as thyroidectomy performed in many of patients with microcarcinoma, there were two main reasons. Firstly, many of our patients declared acceptence for only one-step surgery, and many of them refused potential reoperation in case of some post-surgical indications, like for example lymph nodes metastasis, multifocal disease or positive margins. Secondly, more common one was, that majotity of our patients had microcarcinoma in multinodular goiter. So, in these cases we performed thyroidectomies.
Ad. 2 Dear Reviewer, indeed, the practice of UG-FNAB performing in such small tumors has changed, but in our opinion, only due to indirect reasons, i.e. because of the events described in the study (active surveillance possiblity, personalized medicine introduction or pandemic time). These events caused, that many physicians avoided UG-FNAB in small nodules, what we observed since 2016. After this point the incidence trend of PTC started to drop. However, in the cases of patients with a tumor of 5mm in diameter or even less, and with EU-TIRADS-PL 5, we decided to perform UG-FNAB.
Ad. 3 Dear Reviewer, thank you very much for this remark. Indeed, we are aware of the fact, that in our study, we did not analyze the patients with AS. We included to the study only the patients with available histopathology results. We know, that after 2017 when this “therapeutic” option was highlighted and promoted, many of our potential patients were observed, and not sent to surgery. And it was the potential reason (indirect) of the lower number of TC after 2017. One of the inclusion criteria of this study was obtaining histopathology results, so the study included selection bias because we evaluated only patients with histopathology, so who underwent surgery. The histopathology results of all patients were mandatory for this study to form any conclusions. We included this information in the “Limitations of the study.”
Ad. 4 Indeed, total thyroid resection was performed more often in PTC than in other types of TC. Such situation was observed, because of the fact, that technically in many cases of PTC we were able to perform radical procedure in this malignancy than in other types like even follicular TC, medullary TC, undifferentiated TC, lymhoma, sarcoma or squamous cell carcinoma. We added the sentence explaining this observation to the text of the manusript. “In majority of PTC cases we were able to perform radical surgeries, whereas in the other types of TC like medullary, undifferentiated, sarcomas, squamous cell carcinomas or even follicular TCs, sometimes we were not. In many of these last cases reoperation had to be performed.”
We believe that the findings of our study will interest physicians and researchers who would like to address changes in thyroid cancer management.
The type of submitted manuscript is an original article, and the style and format have been prepared according to submission guidelines.
All authors have read the amendments and agreed to publication.
Thank you for reviewing our manuscript and considering it for potential publication. We appreciate your time and look forward to your response.
Kind regards,
Krzysztof Kaliszewski MD, Professor

Reviewer 2 Report
The study by Kaliszewski and co-workers analyses the incidence trend and management of thyroid cancer. The manuscript is interesting, however, it cannot be accepted in the current form. Suggested changed are listed below.
[1] In the Abstract, it is worth noting from which region of the world/country the thyroid cancer patients concerned by the analysis are from.
[2] It is worth mentioning that in 2022 WHO introduced a new classification of thyroid cancers neoplasms (e.g., Endocr Pathol. 2022 Mar;33(1):27-63).
[3] Please explain why the manuscript omits the topic of kinase inhibitor therapy in thyroid cancer, e.g., dabrafenib and trametinib for treatment of BRAF-mutated anaplastic thyroid cancer or selpercatinib for the treatment of thyroid cancers with RET gene mutations or fusions.
[4] Please explain the abbreviation “UG-FNAB” and “PTMC AS” in the legends to Figure 1 and 2, respectively.
[5] Line 351: “BRAF” is a gene symbol, therefore it should be italicized. Similarly, please correctly write the names of other genes that are listed in line 424.
[6] In line 446 please add the following reference: Int J Mol Sci. 2021 Oct 31;22(21):11829.
The study by Kaliszewski and co-workers analyses the incidence trend and management of thyroid cancer. The manuscript is interesting, however, it cannot be accepted in the current form. Suggested changed are listed below.
[1] In the Abstract, it is worth noting from which region of the world/country the thyroid cancer patients concerned by the analysis are from.
[2] It is worth mentioning that in 2022 WHO introduced a new classification of thyroid cancers neoplasms (e.g., Endocr Pathol. 2022 Mar;33(1):27-63).
[3] Please explain why the manuscript omits the topic of kinase inhibitor therapy in thyroid cancer, e.g., dabrafenib and trametinib for treatment of BRAF-mutated anaplastic thyroid cancer or selpercatinib for the treatment of thyroid cancers with RET gene mutations or fusions.
[4] Please explain the abbreviation “UG-FNAB” and “PTMC AS” in the legends to Figure 1 and 2, respectively.
[5] Line 351: “BRAF” is a gene symbol, therefore it should be italicized. Similarly, please correctly write the names of other genes that are listed in line 424.
[6] In line 446 please add the following reference: Int J Mol Sci. 2021 Oct 31;22(21):11829.
Author Response
Journal: Cancers September 30, 2023
MDPI
Special Issue: “Classification, Risk Assessment and Clinical Management of Malignant Nodules.”
Guest Editors: Prof. Claudio Spinelli, Dr Marco Ghionzoli
Dear Editors and Reviewers,
At the very beginning we would like to thank you very much for the possibility to re-submit our revised manuscript entitled “The Incidence Trend and Management of Thyroid Cancer – What has Changed in the Last Years: Own Experience and Literature Review.” for consideration as an original article in Special Issue “Classification, Risk Assessment and Clinical Management of Malignant Nodules.” Thank you very much for considering it for potential publication in Cancers.
We would like to thank you for the very thorough reviews and for the advices and constructive criticism, which have been valuable for improving our paper. All of the suggestions for changes and improvements were very helpful to us, and we have revised the manuscript according to the recommendations made in the reviews. All of the changed, deleted and added portions of the manuscript are marked by using Track Changes. According to the reviewers’ instructions we corrected our manuscript point-by-point as follows:
Reviewer 2
At the very beginning, we would like to thank you very much for your opinion, that this is “interesting manuscript”. Thank you very much. As far as your comments, we answered them point by point as follows:
Ad. 1 We added to the abstract the information about the region of the world, in which the analysis was performed. Thank you. “We analyzed patients treated in a single surgical center in eastern Europe (Poland).”
Ad. 2 We mentioned in the manuscript, that in 2022 WHO introduced a new classification of thyroid neoplasms. “In 2022 WHO introduced the 5th Edition of the Classification of Endocrine and Neuroendocrine Tumors, which is related to the thyroid gland. Baloch ZW, Asa SL, Barletta JA, Ghossein RA, Juhlin CC, Jung CK, LiVolsi VA, Papotti MG, Sobrinho-Simões M, Tallini G, Mete O. Overview of the 2022 WHO Classification of Thyroid Neoplasms. Endocr Pathol. 2022 Mar;33(1):27-63. doi: 10.1007/s12022-022-09707-3. One of the most important and newest classification change is division of thyroid tumors for BRAF-like malignancies represented by PTC with many morphological subtypes and RAS-like malignancies represented by invasive encapsulated follicular variant of PTC and FTC.
Ad 3. Dear Professor, thank you very much for this special question. The paper was written and comes from surgical center, in which such highly professional treatment is not performed. We know, that in this field of oncological treatment, especially in cases, where surgical methods are insufficient or can not be introduced, many changes are observed. However, because the authors do not treat the patients with thyroid neoplasms using these methods, we decided not to take this topic. We decided to describe more accurately only these clinical issues, which relates directly to our clinical management. Moreover, including this topic in the manuscript would lengthen the discussion section, which, as reviewer 1 suggested, is too long.
Ad 4 We explained the abbreviation of UG-FNAB and PTMC AS in the legends to Figure 1 and 2. Thank you for this remark.
Ad. 5 We italicized genes names in the whole manuscript. Thank you for this remark.
Ad. 6 We added reference Ratajczak M, Gaweł D, Godlewska M. Novel Inhibitor-Based Therapies for Thyroid Cancer-An Update. Int J Mol Sci. 2021 Oct 31;22(21):11829. doi: 10.3390/ijms222111829. in line 446. Thank you.
We believe that the findings of our study will interest physicians and researchers who would like to address changes in thyroid cancer management.
The type of submitted manuscript is an original article, and the style and format have been prepared according to submission guidelines.
All authors have read the amendments and agreed to publication.
Thank you for reviewing our manuscript and considering it for potential publication. We appreciate your time and look forward to your response.
Kind regards,
Krzysztof Kaliszewski MD, Professor

Round 2
Reviewer 1 Report
The previous suggestions to improve the manuscript were appropriately adressed and the questions were answers by the authors. I have no further concerns at this point.
No major changes needed